# Assessing capacities to strengthen intersectoral collaboration in Territorial Public Health Councils in the Republic of Moldova

Oleg Lozan[1], Valentin Mîța[1], Daniela Demișcan[1], Tatjana Buzeti[2], Peter Beznec [3], Valeriu Sava[4], Ala Curteanu[5,6], Constantin Rîmiș [5], Robert Canavan[7,8], Helen Prytherch [7,8]*

1 School of Public Health Management, Nicolae Testemitanu State University of Medicine and Pharmacy, Chișinău, Republic of Moldova, 2 WHO European Office for Investment for Health and Development, Venice, Italy, 3 The Centre for Health and Development Murska Sobota (CHD), Murska Sobota, Slovenia, 4 Swiss Agency for Development and Cooperation (SDC), Chisinau, Republic of Moldova, 5 Healthy Life Project: Reducing the Burden of Non-Communicable Diseases in Moldova, Chisinau, Republic of Moldova, 6 Mother and Child Institute, Chisinau, Republic of Moldova, 7 Swiss Tropical and Public Health Institute, Allschwil, Switzerland, 8 University of Basel, Basel, Switzerland

* helen.prytherch@swisstph.ch

**Data Availability Statement:** All relevant data are within the manuscript and its Supporting information files.

## Abstract

### Background

The government of the Republic of Moldova, endorsed the principle of Health in All Policies (HiAP) through its health sector reforms to address the rising burden of non-communicable diseases and health inequalities. Territorial Public Health Councils (TPHCs) were created to ensure the coordination and management of the population's health in their respective district. This study assessed the capacities of the TPHCs to identify areas in need of support for strengthening their intersectoral collaboration role in public health at local level.

### Methods

A mixed-method approach, using qualitative and quantitative techniques, was used to compare the perceptions of all TPHC members (n = 112) and invitees (n = 53) to council meetings from 10 districts covering all geographical areas of Moldova. The quantitative information was obtained using a cross-sectional survey, while the qualitative aspects were assessed within focus group discussions (FGDs).

### Results

Half of all TPHC members, including 75% from groups with a non-medical background, did not attend a public health course within the last three years. Overall, groups with a medical background were more aware of the legislation that governs TPHC activity and intersectoral collaboration compared with those with a non-medical background. The FGDs of TPHC meetings revealed that members had an insufficient level of understanding of intersectoral collaboration to solve public health issues and lacked clarity about their place and role within the TPHC.

**Funding:** The assessment was conducted as part of the implementation of the Healthy Life Project to Reduce the Burden of Noncommunicable Diseases in the Republic of Moldova (Phase 1: 2017-2021) that was funded by the Swiss Agency for Development and Cooperation (SDC). The authors received no specific funding for the work.

**Competing interests:** The authors have declared that no competing interests exist.

**Abbreviations:** FGDs, Focus group discussions; HiAP, Health in all policies; MHLSP, Ministry of health, labour and social protection; SDC, Swiss Agency for Development and Cooperation; SDG, Sustainable development goal; SUMP, State University of Medicine and Pharmacy; Swiss TPH, Swiss Tropical and Public Health Institute; TPHC, Territorial public health council; UHC, Universal healthcare.

## Conclusions

HiAP implementation was found to be suboptimal with insufficient capacity at local level. TPHC members' ability to deal with public health issues were severely impaired by a general lack of knowledge and understanding of how to utilize the TPHC platform for maximum benefit. Reforming TPHC regulation is required in addition to extensive capacity building for TPHC members to increase member understanding of their roles as intended by TPHC regulations, including the facilitation of intersectoral collaborations.

## Background

Improving communities' health, health equity and committing to the objectives set by the United Nations Sustainable Development Goals (SDGs) and universal health coverage (UHC) is strategically complex and requires considerable effort and resources to promote, achieve and maintain [1]. Many of the targets set in the health and well-being SDG 3 are unattainable solely by the health sector, let alone the 16 other SDGs that also directly or indirectly impact health [2, 3], thus, require a multisectoral approach to achieve these common goals. The unintended consequences of non-medical circumstances (e.g., a person's schooling, living and working environment) are addressed in the SDGs and are widely recognised as having an impact on a population's health [4]. These circumstances, known as the social determinants of health, include factors, such as access to education, good nutrition, reliable sanitation and economic opportunities [5, 6].

Although multisectoral collaboration initiatives for equity in health were mandated by some governments in the latter part of the 20th century [7], the Health in All Policies (HiAP) concept was only introduced in 2006 by the Finnish Presidency of the European Union, it was endorsed and promoted by the World Health Organization (WHO) and encourages government agencies to generate public policies by collaborating across sectors horizontally and vertically, especially where health is not the foremost consideration. The HiAP approach systematically takes into account the health implications of decisions, seeks synergies, and avoids harmful health impacts in order to improve population health and health equity [4, 8, 9]. Albeit HiAP and other initiatives are increasing globally to tackle inequity in health systems, they are still in their infancy and most countries that have implemented them have done so after the year 2000 [10]. Various levels of government engagement (national, regional, and local) in the collaboration process are required to address the many social determinants of health to implement HiAP, with the main objective of forming long-term governmental policies. Despite that, only a small number of studies have been done focussing on local level (city/municipal) implementation [7, 11, 12].

In challenging the growing non-communicable diseases burden, related to lifestyle and behavioural factors, and reducing health inequalities and achieving UHC, the Republic of Moldova (hereinafter Moldova), endorsed the principle of HiAP, through its health sector reform process implemented by the Ministry of Health with support from the WHO Country Office [13–17].

In 2001, the responsibility for public health was decentralised and public health centres, that were once concentrated in regions, were established and their administration assigned to 35 districts and two municipalities in the country [14, 18]. Although this transfer of power was enshrined in law in 2009, it was not until 2016 that the Ministry of Health, Labour and Social

Protection (MHLSP) created advisory councils for public health in each district, under the auspices of their corresponding Public Health Centres, to ensure the effective coordination and management of health of the population in their respective districts [18, 19]. These devolved Territorial Public Health Councils (TPCH), which are subordinate to the MHLSP, comprise representatives from health institutions, the education sector, social protection sector and local authorities and are chaired by the heads of the Territorial Centres of Public Health; they were mandated to: (i) examine the organisation and functioning of the district's health system; (ii) promote priority public health objectives; (iii) coordinate the activities of medical and pharmaceutical facilities; (iv) ensure the coordinated implementation of legislative and normative acts and of the national health programme; (v) improve the quality of health care; and (vi) improve health outcomes for the population [18].

Although TPHCs are now firmly established and functioning, a broader understanding of the TPHC mandate and capacities could benefit all factions involved and thereby strengthen their role in regard to the continuing health and public health care challenges they face [15, 20].

The Moldovan government intends to strengthen the role of the TPHC in disease prevention, health promotion, risk reduction, governance and intersectoral cooperation, in an effort to progress its health systems strengthening reforms with the aim of reducing health inequalities and achieving its 2030 UHC and health related SDGs [14, 21]. The "Healthy Life Project to Reduce the Burden of Non Communicable Diseases," funded by the Swiss Agency for Development and Cooperation (SDC) and implemented by the Swiss Tropical and Public Health Institute (Swiss TPH), in partnership with the School of Public Health Management, WHO European Office for Investment for Health and Development and the Murska Sobota Centre for Health and Development, Slovenia, was introduced to Moldova to further these aims and in particular reduce the burden of non-communicable diseases, with a focus on rural areas. In 2018, the Slovenian partners in the project proposed to develop and pilot an assessment methodology that the Moldovan project team could apply together with the TPHCs to identify areas in need of support and fulfil their mandate of promoting health in all sectors. The present study comprehensively assessed the capacities of the TPHCs with the aim of identifying areas in need of support for strengthening their intersectoral collaboration role in public health at the local level.

## Methods

### Study overview

A mixed-method approach, based on qualitative and quantitative techniques, was carried out to compare the perceptions of all members with the right to vote within 10 TPHCs from 10 districts in all areas of Moldova. Recruitment of informants and data collection took place from 16 to 24 May 2019. A representative sample was created through a probabilistic method—three districts were randomly selected from each geographical area of the country (i.e., North—Briceni, Edineț, Fălești; Centre—Ungheni, Orhei, Criuleni and South—Ștefan Vodă, Cahul, Taraclia) and one district of UTA Gagauzia (Vulcănești).

The quantitative information was obtained using a cross-sectional survey, while the qualitative aspects were assessed within focus group discussions (FGDs).

All members of the TPHC with voting rights appointed by the MHLSP Order no. 963 from 17 August 2018 "on approval of the nominal composition of the Territorial Public Health Councils," from 10 districts (n = 112), were included in the study in addition to the participants invited to the TPHC meetings with no voting rights (n = 53). The final group of participants was composed of four categories: (i) members who were part of the public health state

monitoring system (e.g. TPHC presidents–territorial heads of public health centres–and their secretaries; n = 26); (ii) professionals in the public health field (e.g. medical employees from district hospitals, healthcare centres, district emergency and healthcare substations; n = 71); (iii) professionals who contribute indirectly to public health (e.g. representatives of the Civic Protection and Exceptional Situation Service and the Territorial Education Department; n = 15); and (iv) representatives of the district council from the territory (e.g. presidents/vice presidents from the district; n = 7). The four categories of participants included representatives of the following institutions/organizations: 1) District Council (Chairman or Vice-chairman); 2) District Education Division; 3) Local Public Authorities (mayors); 4) district public medical-sanitary institutions (heads of public health centres, hospitals, primary healthcare institutions, Emergency Medical Assistance station); 5) private medical institutions (dental, pharmaceutical); 6) social assistance; 7) police; 8) fire service; 9) Service of Civil Protection and Exceptional Situations; 10) District Trade Union Committee of the Trade Unions Federation "Health"; 11) Territorial Agency of the National Health Insurance Company; 12) local non-governmental organizations.

## Framework

The United Nations Development Programme (UNDP) Capacity Assessment Framework [22] was adapted to conduct a capacity assessment of the TPHC using its three common dimensions:

1. the points of entry and levels of capacity dimension, which were determined as (i) enabling environment level (systems and frameworks in place or needed to formulate/implement policies and strategies); (ii) organizational level (resources in place i.e., human resources physical resources, intellectual resources etc. and organizational or managerial methods used); and (iii) individual level (knowledge, abilities, values, personal attitudes).

2. the core issues dimension, which analysed issues such as: (i) presence of institutional arrangements; (ii) leadership; (iii) level of knowledge in the field and understanding of public health issues and developing solutions; understanding the accountability and (iv) obligations by TPHC members; and

3. the technical and functional capacities dimension, assessing technical (associated with particular areas of expertise and practice) and functional capacities of the TPHC which included: (i) identifying the degree of engagement of stakeholders, partnerships and collaborative mechanisms in place, engagement of civil society and the private sector in handling and solving public health issues at territorial/district level; (ii) analysing the capacity to access, gather and synthesize data to define a vision; (iii) determine the capacity to set priorities, formulate objectives, policies and strategies, (iv) capacity in place to prepare, manage and implement human and financial resources; and develop a proper monitoring process; (v) capacity to evaluate and measure results to adjust policies, codify lessons and ensure accountability.

## Cross-sectional survey

A cross-sectional study was carried out to compare perceptions of TPHC members and those invited to TPHC meetings. Persons that refused to take part in the study and those that were absent at the time of the assessment were excluded from the study. The age of the respondents and the number of years of their work experience were recorded and detailed in Figs 1 and 2 in S1 File.

Respondents were offered the choice of a standard face-to-face interview to complete paper questionnaires or they could complete the questionnaires themselves electronically. In total, 165 questionnaires were collected, 82 of which were collected from council members and 83 from other public health officials invited to council meetings. There were a total number of 82 questionnaires validated for analysis. The questionnaire included 100 questions organised into four basic components: general data, individual capacity level data, organizational capacity level data and environmental capacity level data (see S2 File). The data were collected between 16 May and 24 May 2019.

## Focus group discussions

FGDs were conducted to assess the attitudes, behaviour, opinions, beliefs, knowledge and abilities about specific issues from a well-informed group invested with authority, in this case the TPHC members and the invitees. There were 10 focus groups held at TPHC meetings and they were composed of the same participants as in the cross-sectional study.

The group discussion research tool was composed of an introductory and seven thematic blocks of questions, including explanatory questions (see S3 File). These were as follows: (i) assessing the level of knowledge and skills in the field about population's health in the district; (ii) Identifying Stakeholder Engagement, Partnerships, and Collaborative Mechanisms; (iii) Understanding the role of TPHC, accountability and obligations by members; (iv) institutional arrangements and leadership; (v) Council's capabilities to collect and synthesize data to create a vision; (vi) determining the ability to formulate objectives and set priorities, manage human and financial resources; and (vii) the capacity to develop an appropriate policy monitoring and adjustment process.

The interviews were led by the moderator; two other researchers participated as observers while monitoring the non-verbal behaviour of the participants. Interviews were recorded, with the agreement of the participants, and subsequently manually transcribed and coded.

## Consent

Informed consent to participate was obtained verbally at the beginning of the interview from each of the study participants. Study participants were informed in both the qualitative and quantitative parts of the study that they could withdraw from the study at any time without harm or that they could refuse to answer certain questions if they felt uncomfortable. Participation in the study was voluntary and unpaid. Persons who refused to take part in the study and those who were absent at the time of the assessment were excluded from the collection of data (did not participate).

## Processing and interpretation of collected data

Quantitative analysis was carried out by coding and processing the data using Microsoft Excel 2013, the results obtained being presented in the form of mean value with standard deviation (SD) or proportion (%), depending on the type of variable examined. In the data processing process, the hypotheses were confirmed/disproved in relation to the results obtained. The statistical significance of the differences revealed in the responses provided by the participants was determined by Student's t-test or chi-square test. Completed questionnaires with >20% missing data were excluded from the analysis.

A phenomenological analysis was carried out to highlight the major topics emerging from the transcripts, which allowed the representation of the various aspects and perspectives of the people who participated in the study. This method facilitated the understanding of how personal priorities, organizational and environmental conditions influenced the beliefs,

aspirations and capabilities of study participants. The data analysis consisted of examining, classifying data by categories, tabulating and recombining the observations during group interviews. The transcription of the verbal narratives was carried out following the initial coding, revealing the thematic relationships, grouping the themes into clusters and searching for themes that reflect the common experiences of those included in the study. Fragmented texts were integrated for a holistic view of the data. Reflexive analysis was used to formulate the conclusions.

## Ethics approval

The data were collected according to MoH Provision no. 98d of March 28, 2019, "Regarding the evaluation of the activity of the Territorial Public Health Councils", during the ordinary meetings. It should be noted that at the time of the study, approval by the Ethics Committee was not mandatory for this type of research. This procedure was carried out and verbalized at the level of two research forums, in which the study was subjected to debates regarding the organizational, ethical and methodological aspects (quantitative and qualitative) of the research, and was later approved by all the Forum participants, including the teaching staff.

## Results

### Groups and views on intersectoral activity

The majority of participants (57%) were of the opinion that there were other mechanisms of intersectoral collaboration in the field of public health at district level prior to the establishment of the TPHC (Fig 1). However, almost half of members (43%) from group IV (district presidents/vice-presidents) stated that they knew nothing about other mechanisms of collaboration before the implementation of the TPHC, a third of members of group III declared the same.

### Level of knowledge and understanding of public health problems

TPHC includes people that have no medical background; thus, it was necessary to ascertain the members' level of commitment to gain knowledge in the field of public health by their

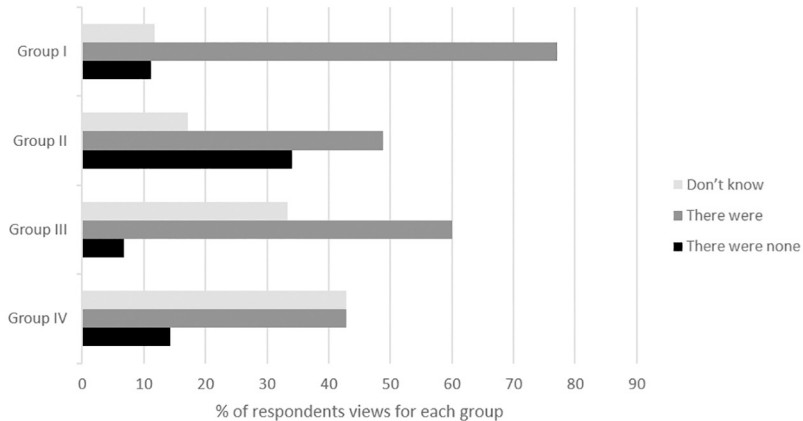

**Fig 1. Respondents' perceptions of the existence of other mechanisms of intersectoral activity in the public health field at district level prior to the implementation of the TPHC.** Group I, members of public health state monitoring system; Group II, public health professionals; Group III, professionals who contribute indirectly to public health; Group IV, representatives of the district councils.

attendance of public health courses within the last three years. Two thirds of members from group I (67%) and 32% of members from group II (both groups with a medical background) considered that they did benefit from public health training opportunities within that time frame. Of the members from groups with non-medical backgrounds (group III and IV) 20% and 43%, respectively, perceived a benefit from the courses undertaken.

Of the 67% of respondents that said that they had undergone public health training in the last three years, approximately two thirds (the majority being in group IV) had undertaken public health training courses promoted by the "Healthy Life" Project between October and November 2018 (Fig 2). Respondents claimed to have attended multiple courses in group I (50%) and group II (46%) promoted by the MHLSP, continuing education courses promoted by the State University of Medicine and Pharmacy "Nicolae Testemitanu" and courses promoted by the "Healthy Life" Project.

Three out of four members from groups III and IV (non-medical backgrounds) did not attend any courses in public health within the last three years. The most often cited reason was the lack of time (38%), distance from the course venue (19%) and that their participation was restricted by their managers. Frequently cited reasons for groups I and II (doctors) for not attending courses within the last three years were the lack of time (30%), lack of influence on their salary (15%) and participation was restricted by their managers (10%) (see Fig 3 in S1 File).

Respondents were asked to self-assess their theoretical and practical knowledge of public health issues to effectively operate within the TPHC (Fig 3). More than 70% of respondents from group III considered that they required additional knowledge in certain areas, compared to less than 50% of members in all other groups. None of the members in either of the groups suggested that they did not need to be informed because it was not their responsibility or that they were not interested in gaining further knowledge.

In the FGDs with TPHC members, when questioned about public health issues regarded as a priority for their district, the majority relied solely on their own opinions and experiences without reference to any reports, statistics or data, as illustrated by the following comment in an FGD:

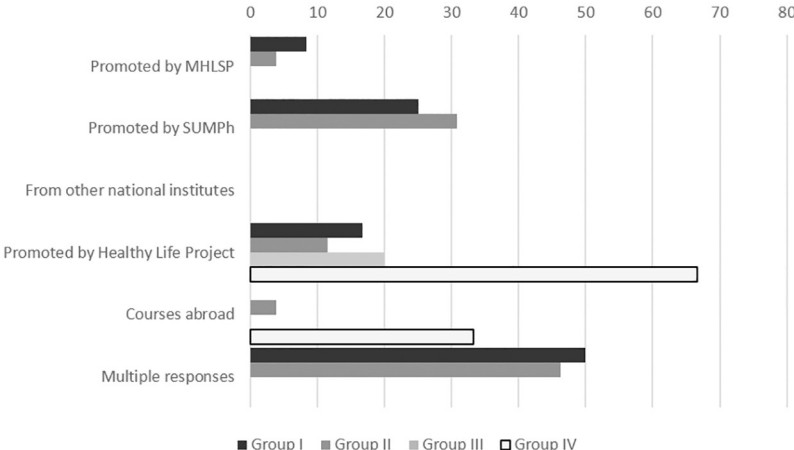

**Fig 2. The various public health courses taken by respondents within the last three years, stratified by groups (%).**
Group I, members of public health state monitoring system; Group II, public health professionals; Group III, professionals who contribute indirectly to public health; Group IV, representatives of the district councils; MHLSP, Ministry of Health, Labour and Social Protection; SUMPh, State University of Medicine and Pharmacy "Nicolae Testemitanu".

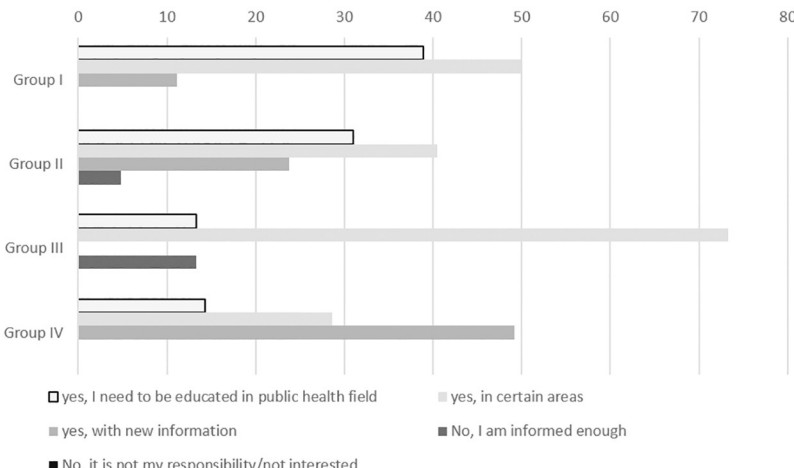

**Fig 3. Percentage of respondents' opinions on their need for knowledge in the field of public health.**

*"I say that one of the most serious problems is diabetes. I have this disease myself and I understand how tough it is!"*

*[representative of the district council in FGD7]*

Further, some council members appeared not to properly understand the notion of a public health issue or the need for a multisectoral approach to it. In one FGD, an example was given of prevention of a sedentary lifestyle by organising cycling tracks in the locality. A mayor participant responded:

*"It is not a mayor's problem the physical activity of the population. It's just everyone's responsibility to do sports, go to gyms, or I don't know what. . .Everyone is responsible for their own health! I will make cycling tracks and then I have trouble with the road police?!"*

*[FGD9]*

### Presence of institutional arrangements and leadership

The legislative Act, established in 2016, that governs TPHC activity and provides for institutional intersectoral collaboration in the public health sphere was perceived by its members differently. The respondents from groups I and II were most aware of the legislative Act that regulates TPHC activity, 88% and 48% respectively. The same was true for 13% of respondents in group III and 29% in group IV. The importance of an insufficient legislative-normative framework as a barrier to the activity of TPHC was assessed with an average score of 7.0 (where the minimal importance = 0 and maximal = 10) (see Fig 4 in S1 File).

Although the majority of respondents (77%) acknowledged that there was an approved TPHC activity plan and had read it, 11% of respondents had not read it and 11% knew nothing about such a plan. Furthermore, 16% of members did not regard the council as a platform for intersectoral activity at the district level for prevention of diseases and health promotion, social determinants of health and/or promotion of the health in other policies.

For the purpose of increasing the efficiency of TPHC activity, members were asked if the current TPHC members represent all the authorities that are responsible for public health in

**Table 1. Characterisation of relations with colleagues within TPHC (N = 82).**

| Characterisation | No | % |
|---|---|---|
| Teamwork | 24 | 29.3 |
| Collaboration among colleagues | 32 | 39 |
| Good but with minor misunderstandings | 8 | 9.8 |
| Tense | 1 | 1.2 |
| Conflictual | 1 | 1.2 |
| Indifferent | 1 | 1.2 |
| Teamwork plus collaboration among colleagues | 12 | 14.6 |
| No reply | 1 | 1.2 |

the territory. The majority (65%) of members stated they did not and that the composition needed to be changed so as to include representatives from other authorities/parts of the society in the district, such as the representatives of the Ministry of Internal Affairs, National Agency for Food Safety, local non-governmental organisations, Ecology Service, religious cults, local mass-media, and the district social assistance. A third (35%) considered that members did represent all authorities responsible for public health in the territory.

The opinions of TPHC members were sought on the existence of persons in the position of leaders (regardless of sector) in the field of public health promotion. The majority (78%) considered that there were leaders, while 17% were unsure. The perception of relations within TPHC were sought and characterised differently by members (Table 1). The majority of members' (39%) relations within the council were characterised as one of collaboration, in contrast, 4% of respondents stated that relations within the TPHC were tense, conflicted and indifferent.

Public Health Centre representatives were asked if the TPHC management had a level-headed and efficient approach, 89% of the respondents thought that they did. The opinions expressed by TPHC members were divided regarding the organization of intersectoral actions in the field of public health at territory level. Between 60 and 70% of respondents from group I and II were of the view that the National Agency for Public Health was the public authority responsible for carrying out intersectoral actions in the field of public health, 40% of group III were of the same opinion but almost a third of group III also considered the health centre responsible (Fig 4).

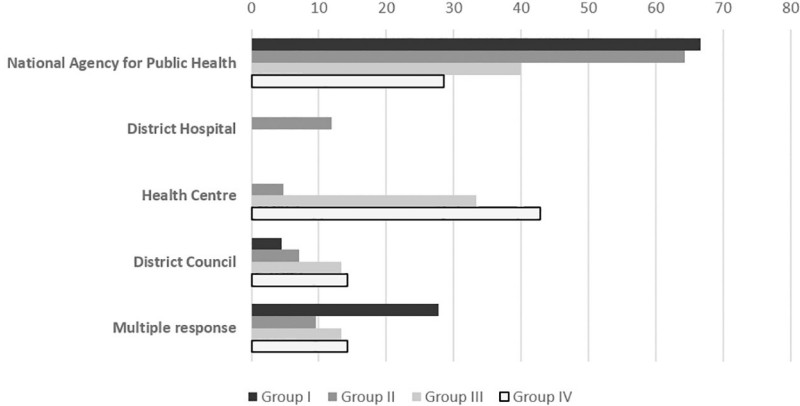

**Fig 4. Distribution of representatives' opinions regarding the organization for intersectoral activities in the field of public health.** Group I, members of public health state monitoring system; Group II, public health professionals; Group III, professionals who contribute indirectly to public health; Group IV, representatives of the district councils.

Among the multiple responses given by all members, a common theme regarding joint management of the TPHC stood out. This was supported by members' opinions regarding how the management of TPHC should be organized. Only 57 of 82 members (70%) took part and their opinions were divided; 26% thought that the management should be elected by TPHC vote, 19% considered that the management should be by order of rotation TPHC members, a further 19% thought that it should be managed by the National Agency for Public Health and 16% considered the local authority should be responsible.

Since council's Regulation is approved only by Order of the MHLSP, it appeared that public health problems were issues limited to the health system only. In many districts, officials of the Public Health Centres reiterated that it is was complicated to convene TPHCs because they had no means to make people from non-medical fields accountable. A secretary from a council in FG3 revealed:

> "It's really hard to bring them all together...I had cases when I kept calling the lady from the education department and she told me she had no time to come. I reproached that she is in the list approved by the ministry. And? What can I do if she does not come? She is not subject to the orders of the Minister of Health..."
>
> [council secretary in FGD3]

Accordingly, most councils suggested that their work should be done on the basis of a much more powerful act than a ministerial order, such as a government decision or a law be passed that would oblige all those in charge of decision-making in all sectors to have the health component in their action program with the obligation to perform the tasks.

> "It is necessary to change the legal status of these councils. More power needs to be given for the decisions taken! In this council, there must be representatives of the prosecutor's office by all means! We need lawyers!"
>
> [TPHC chairman in FGD6].

Observations during some TPHC meetings revealed that not all territories were in similar situations; the councils were organized and led very differently, which appeared to create confusion and various interpretations among members. Thus, although the council Regulation stipulated that the TPHC is not subordinate to the local public authorities [13, 19], during meetings, it was observed that the meetings took on more of an authoritarian character shifting into a reporting session to the chair of the district council. It was also apparent that TPHC members did not know the legal health obligations (such as people's access to medical services and confidentiality of medical data).

Council members, and in particular the representatives of the Public Health Centres, frequently reported that "vertical communication" was unsatisfactory and vague, (i.e., relationships with the higher public authorities, the National Public Health Agency and the MHLSP). Further, there was no support offered by these authorities, reporting to them was a formality and no feedback was ever given. One TPHC chairman reported:

> "Last year we met every month, as required...At the end of the year we sent the annual report, but we did not get anything back, either that it's okay or that it's bad...This year we meet every three months...So, we were really hurt, if no one appreciates our perseverance..."
>
> [FGD9]

It was also observed that the frequency of TPHC meetings differed among districts, some districts being monthly and in others twice per year.

Furthermore, in some districts serious problems in finalising the nomination procedure for TPHC members was found due to the approval of their list being held up at MHLSP for several months in a row. Thus, for those councils affected, it was difficult to issue any decisions, since the participants in the meeting were not legally nominated yet for their respective responsibility. In the majority of meetings, a much higher degree of power to punish (fine) those who were not responsible for their own health was expressed. One respondent stated:

*"We should be allowed to be able to punish those who do not listen and are irresponsible through unhealthy behaviours. . .For example, those who smoke, drink or do not want to get tuberculosis treatment!"*

*[FGD7]*

### TPHC members' perception of responsibilities and obligations

TPHC members' perception of the importance of each TPHC member was sought (see S4 File for more details). Respondents considered that the TPHC president and the secretary were the most important members from a list of 19, both with a score of 9.2 (0 = no importance; 10 = maximum importance), followed by the representative of the district council with a score of 9. The least important role, according to the opinion of respondents, was the representative of religious cults (score of 5.5) (For more details see S5 File).

As to whether the participants agreed with the formal character of TPHC activities, they rated an average score of 4.1 (0 = total disagreement; 10 = total agreement). To the question, "How much does TPHC activity influence the motivation (including financial) of TPHC members?" Members assigned an average of 6.4 (0 = very little; 10 = very much). Further, when questioned if they considered that the quality of TPHC's work is influenced by the fact that decisions are only advisory in nature, participants assigned an average score of 7 (0 = very little; 10 = very much).

During FGDs, participants were asked about how they felt when they were informed that they were going to become TPHC members; many acknowledged that they felt "assigned an extra responsibility" or even "a burden." The majority stated that they were very busy and did not have the time to "sit in meetings."

In several districts, it was observed during TPHC meetings that the chair of the district council and the mayor of the district were present only at the beginning, they then signed the necessary documents and withdrew explaining that they were very busy. In addition, in some other districts, it was noted that people were present at the meeting to stand in for the nominated members to confirm their presence but who did not understand the role and essence of the TPHC's work. In some districts, the TPHCs were regarded as extremely formal and their activities were not considered to have any significant local impact, indeed, some participants admitted (people from non-medical areas in particular) that they did not know or understand their role and obligations in the council.

It was often stated during FGDs that the responsibility for the health of the population falls directly on the workers in the health system and the population themselves. Further, whilst observing FGDs, it became clear that representatives, e.g., of local authorities, the education department and media, did not understand their role and responsibilities in solving public health problems. It was also observed that the majority of TPHC decisions contained

directions to health workers without distributing responsibilities to representatives of other structures outside the medical sphere. In some districts, during TPHC meetings, it appeared that they were called solely to listen to reports from representatives of health institutions and to designate some tasks that were, in fact, already a part of their job duties and the essence of their professional activity. (e.g., in one council decision, family doctors were tasked "to monitor the health of patients with chronic diseases", which was already part of the performance indicators in primary care). Several council secretaries also appeared to be unaware of their responsibilities, a reoccurring theme during FGDs was that it was difficult for them to make members accountable for a structure that is essentially only "consultative" by its nature.

Two TPHCs members in FGDs had the opinion that the democratically elected Mayor could not be compelled to act on a TPHC decision, further suggesting that the decision may not even be read, especially if finances were short.

### Technical and functional capacities of the TPHC

**TPHC capabilities to access, gather, and synthesize data to define a vision and mandate.** The TPHCs readiness for performance of duties were assessed (Table 2) by the TPHC members on a scale of 0–10 (0 = missing; 10 = sufficient) (see S4 File for more details). The TPHC gave an average of 7.2 with regards to the provision of information and materials needed to perform their tasks properly. Members assessed the TPHC capacity to draft policies in the field of public health and to establish priorities with an average score of 7.7 from 10. Communication between members of the TPHC was evaluated, and scored an average mark of 8.5 (from a scale of 0 = min; 10 = max).

It appeared from FGDs that TPHC members, especially those outside the health system, were not familiar with the data requirements for public health. The analysis of the working agendas of these councils shows that formal, very complex and general reports were included,

**Table 2. The Territorial Public Health Council readiness for performance of duties.**

| | Degree of preparation for | Average score |
|---|---|---|
| 1. | Examining the current problems of the organization and operation of the district health system | 7.8 |
| 2. | Ensuring collaboration between medical institutions in the territory | 8.0 |
| 3. | Drafting, coordination and promotion of territorial public health programs and exercise control over their implementation | 7.9 |
| 4. | Tackling the health problems of the population in the territory and their determinants | 7.6 |
| 5. | Ensuring the implementation of national and territorial health programs | 7.7 |
| 6. | Drafting and analysis of health profiles in the territory | 7.7 |
| 7. | Ensuring the implementation of policies in the field of human resources in healthcare | 6.0 |
| 8. | Organization and execution of complex response and control measure in public health emergencies | 7.8 |
| 9. | Involvement of the society in the provision of public health services; | 6.9 |
| 10. | Supporting the development of the plans for ensuring high quality drinking water, air protection, waste removal and sanitation; | 7.2 |
| 11. | Coordinating the allocation of financial resources, including investment, in the field of public health monitoring. | 5.7 |
| 12. | Facilitate the organization of sociological studies on the quality of healthcare services provided in the territory and screening of the population. | 6.5 |
| 13. | Measuring results and collecting feedback for adjusting policies | 6.3 |
| 14. | Ensure accountability of all relevant stakeholders | 7.0 |

(0 = very unprepared; 10 = very prepared)

without clear interventions necessary to influence any of the indicators. Materials on various public health topics, presented to the TPHC by representatives of the districts' public health centres, were analysed and found that they were prepared in a very complicated and formal manner. One participant in a FGD stated:

> *"We are shown so many figures, we get lost in them and we do not understand anything. . .for us morbidity, incidence, prevalence are foreign words. Is it much or little? I cannot say! We should have some information closer to our understanding, not just empty statistics! And to explain to us, what is actually wanted from us"*
>
> *[representative of the education department FGD8]*

Similar situations were reported in many of the FGDs. In one TPHC meeting, it was observed that the representative of the Public Health Centre held a 45-minute presentation, much of the information was irrelevant to the topic of the meeting. The specialist complained, nevertheless, that he had to work for two weeks on the presentation giving up other important activities.

**Setting priorities, objectives, policies and strategies.** The TPHC activity strategy was considered for ensuring sustainable intersectoral collaboration in the field of public health at the territory level (Table 2) (see S4 File for more details). Ensuring collaboration between medical institutions (average score of 8), tackling the health problems of the population in the territory (average score of 7.6) and the self-assessment of the TPHC and operation of the district health system (average score of 7.8) were all considered well prepared for by respondents. In contrast, coordinating the allocation of financial resources was not considered so well prepared for with an average score of 5.7.

Whilst observing many TPHC meetings, it became apparent that there were no clear objectives focused on public health issues, furthermore, the discussions went way beyond the planned agenda.

It was confirmed by some councils in FGDs that the secretary had requested that members propose an agenda to be included in the scheduled one-year session. In other councils, however, they were not so well organised. In the majority of the councils, members representing non-medical institutions could not confirm that they had proposed any topics for discussion in the TPHC. A participant in the discussion stated:

> *"We are only talking about problems here. . .we do not really develop concrete actions. . .We're talking. . .and so what?"*
>
> *[FGD7]*

From the discussions and experiences shared by TPHC members during the FGDs, "diseases and actions necessary to secure their treatment" were discussed. Nevertheless, not much attention was paid to actions to educate the population and promote a healthy lifestyle.

Specific cases were described when council members identified a public health issue during FGDs but did not know how to find a solution, such as the issue of inappropriate menus for child nutrition at pre-school and school level. The problem of children's mental health was also an issue but which, the council members thought, they could not influence in any way.

**Managing human and financial resources.** The lack of sufficient human resources and funds were considered barriers to the activity of the TPHC and the respondents rated this at an average importance level of 7.5. The respondents considered the capacity level of the TPHC, with regards to ensuring the implementation of human resource policies in healthcare,

with an average score of 6 and with regard to budgeting the financial resources, including investment, in public health monitoring, the respondents scored an average of 5.7 (Table 2) (see S4 File for more details).

Both, some chairs of district councils and mayors who participated in FGDs expressed their discontent that they had no employees to deal with the populations' health problems. They tried to explain how overburdened they were with "just as important" issues, so failed to get involved in public health issues, leaving the responsibility for the lack of manpower issues entirely with the health workers.

In many FGDs, participants reported that the lack of available funds either delays solving public health problems or they do not get done at all. Allocating public funds from local authorities to solve public health problems appeared not to be common practice, as explained by a FGD participant:

*"It's results that we gather, inform, know and. . .It stops there?! Furthermore–there, is no money!"*

*[FGD4]*

**Partnerships and collaborative mechanisms in managing public health issues.** To ascertain the efficiency of communication both horizontally and vertically, the respondents assessed the collaboration between various hierarchical levels (see S4 File for more details). The respondents assessed their collaboration with the MHLSP with an average of 5.8 and the National Agency for Public Health with an average of 6.4, local authorities an average of 6.9 and the collaboration and support with foreign sponsors for the TPHC, an average of 4.9.

Respondents also assessed communication barriers faced by the TPHC activity at district and inter-district level with an average score of 6.1 and viewed the TPHC as less prepared regarding mobilising civil society to provide public health services and collectively gave an average score of 6.9. Nevertheless, 60% of members deemed that intersectoral activity at the district level had improved considerably since the establishment of the TPHC and 34% considered that there was a slight improvement. For clarity, the majority (72%) declared that it was the communication and collaboration between medical institutions in the district and the communication between all agencies represented in the TPHC which had improved.

The approach of all councils in FGDs was unambiguous, that public health issues were regarded as the responsibility of medical institutions and physicians, respectively. The task and responsibility for enforcing many councils' decisions was mainly the duty of medical institutions, there were far fewer decisions for other actors involved.

The representatives of medical institutions in the TPHC emphasised that the meetings *"are occasions where they can talk to each other, the doctors, about the problems they have,"* reaching very heated discussions. However, the lack of adequate collaboration between health care institutions was also evident, as illustrated by this statement from a mayor of one village present at the meeting:

*"I have many other problems on my shoulders. Health is the headache of family doctor; she gets salary for this! I have nothing better to do than worry about tuberculosis or other problems! She does not solve mine!"*

*[FGD6]*

Although the importance and necessity of cross-sector collaboration was repeatedly confirmed by the members of the TPHC in FGDs, it was not possible to identify exactly how they saw this collaboration. In contrast, many positive experiences establishing partnerships were identified. In some districts, examples of productive collaboration were given whereby NGOs with external funding were able to contribute to solving specific public health problems.

During several FGDs, TPHC members suggested extending the number of participants in their meetings, in particular, to invite mayors from all villages to get an understanding of how things operate in a district's health sector. Participants were of the view that this would increase the level of collaboration and interest from Local Public Authorities. However, not all members were open to extending the composition of the TPHC, as a chair of a district council stated:

> *"Well, we cannot invite all people in this Council! What shall a policeman do here, for example?"*

> *[district council chair in FGD7]*

A frequently expressed view in FGDs was that clear communication mechanisms should be developed both horizontally (among all actors responsible for different public health issues) and vertically (communication with the governing bodies). Some TPHC members believed that they had too few possibilities to be heard but valuable contributions to be made. It was further alluded to during FGDs that some issues needed to be resolved at a much higher level before finding solutions at a district level.

**Organization of monitoring and policy adjustment.** Meetings of TPHC members are conducted on a monthly basis. Respondents rated the importance of the monthly meetings with an overall average score of 8.1 (see Fig 5 in S1 File). However, the opinion regarding the way these council meetings should be held varied among members and the different groups. The majority of members in all groups (group I– 94%, group II– 64%, group III– 53% and group IV– 71%) considered that TPHC meetings should be held monthly. Others thought that they should be held once every three months (14% in group II and 20% in group III). Some members in groups II (12%) and III (7%) answered that they did not know when the council convened; they would only attend meetings on invitation from the council Secretary/ President.

Another aspect in the monitoring of TPHC activity is the existence of an instrument to assess the council's capacities and approved decisions, only 37% of the respondents had knowledge of such an instrument.

A frequently mentioned issue in FGDs was that the TPHC does not have any legal leverage to monitor the fulfilment of tasks of different actors in the health system, which were set by decisions made during TPHC meetings. A head of the Public Health Centre revealed:

> *"Our decisions are just recommendations. . . .. How can I require a mayor to do something?"*

> *[FGD4]*

Several participants had the view that TPHC decisions should be more transparent and the Council itself should make its work much more transparent.

> *"Information should be open and even promoted with reference to decisions taken at Council meetings. Let's make it clear that this is the issue that has been discussed, these people have to*

*do some things. Let the whole [district] know it! This would perhaps make them more respon-sible for carrying out the activities they are asked for!"*

*[FGD7]*

The implementation of national programs at the territorial level varied between the districts included in the study. In some districts, locally-approved territorial programs with a well-defined action plan and specific partners were determined. Additionally, the costing of actions was done and financing sources identified; sources allocated by the local public authority could also be found. In contrast, some districts were much less clear about the actions envis-aged for implementing the provisions of National Public Health Programs.

## Discussion

This study assessed the capacities of the TPHC with the aim of identifying areas in need of sup-port for strengthening their public health multisectoral collaboration and, as a consequence, their role in disease prevention, health promotion, risk reduction and governance. Regarding the assessment of individual members' public health knowledge; their mere attendance at pub-lic health courses within the last three years, of course, does not imply they were able to digest the necessary information nor to effectively carry out their duties. Nevertheless, it was a basis which was considered measurable for the purposes of this assessment. It was interesting to find that 75% of members from groups III and IV (non-medical backgrounds), that possibly had the least knowledge in public health to be able to effectively operate in the TPHC, did not attend any courses. Although many had their reasons for not attending e.g., no time, distance, managerial restrictions etc only 13% of group III and 14% of group IV considered, in a self-assessment, that they needed to be educated in the public health field to be able to effectively operate in the TPHC. This was compared to 40% of group I and a third of group II that already had a medical background and a familiarity with public health related issues. At first glance it appears that members of group III either felt that they were knowledgeable enough to operate in their capacity within the TPHC framework or were oblivious to the demands of the post. However, further inspection provides that over two thirds of group III indicated otherwise and recognised that they did require extra education, however, only in certain areas. These results appear to suggest that the majority of members were quite willing to engage in courses to effec-tively operate within the TPHC. However, considering only approximately half of all TPHC members attended a course within the three years, other incentives to attend may be required, especially for those that perceived no benefit or did not for their various reasons, as members non-attendance on courses that affect community work can clearly have an impact on the holistic approach of HiAP [23].

Although legislation was implemented by the MHLSP, which governs TPHC activity and provides for intersectoral collaboration, it is concerning that the majority of TPHC members were not aware of it and as a consequence have insufficient knowledge of how the council should operate. This also applies to the approved TPHC activity plan where a fifth of members had not read it and half of those were unaware of its existence. Any legal reforms of the TPHC framework may need to address concerns regarding the convening of TPHC members for the meetings to be useful. A further consideration is when decisions are made, they could be more than advisory to motivate intersectoral collaboration and encourage intersectoral actions.

From discussions with TPHC members, the findings suggested that there was, overall, a low level of knowledge on public health issues. Members generally did not understand the need for a multisectoral approach to solving these issues, mostly relied on anecdotal evidence within meetings and members appeared to have insufficient knowledge of health policy documents

and the issues to be addressed in them. Furthermore, the place and role of members within the council in solving public health problems appeared not to be fully understood and even considered burdensome, this not being unique to the development of intersectoral collaborations in health [24]. The leading figures within the TPHC meetings were by no means immune from criticism as they provided no clear mechanism, direction or guidance for TPHC work. This perception was also assumed by stakeholders unaware of their legal responsibilities in an intersectoral dengue control programme in Mexico [11, 25]. An explanation for this situation could be insufficient public health training that targets the objectives within the TPHC framework. Thus, it appears necessary to develop an extensive training course for TPHC members to understand their roles within the council, as intended by the regulations, and how to progress and overcome public health problems with a focus on horizontal and vertical collaboration.

Political will and public participation are also considered important in facilitating intersectoral actions. While local authorities are in a better position to assess the needs of their community than a national government, media exposure can increase public awareness creating a demand for action and in turn keeping the local authority engaged [7, 11, 12, 26]. Thus, mayors should also have a strong awareness of their roles within the TPHC and how they can advance interventions that benefit the public's health in the long term and their ease for making healthy life choices.

The fact that the composition of the TCPH was largely represented by medical specialists may have reinforced the perception that it was a platform for discussing medical issues, which were considered predominantly the responsibility and for the benefit of the health sector, rather than a unified collaboration of related health topics that could benefit the community at large. In an evaluation of the development and implementation of an intersectoral health policy in a Danish municipality, Varde, non-health sector employees were also concerned that the health sector were dominating their views on health issues and considered that an emphasis on intersectoral collaboration needed to be addressed [24]. There is no doubt that intersectoral cooperation in health can be challenging. This is especially so for those representatives in non-health sectors that may feel out of their depth dealing with public health issues, as was established in the Varde evaluation [24]. Further, the potential frustration of non-health sector members using their limited resources for the benefit of joint public health initiatives, as was also the case in Varde and was resolved by establishing a mutual health fund with the intention of prompting intersectoral solutions [11, 24]. Nevertheless, engaging non-health sector members and incentivising them, by supporting their particular agenda in a health impactful way to achieve unified goals (a win-win strategy), could strongly facilitate the acceptability of intersectoral cooperation [23, 27].

Beyond the UNDP Capacity Assessment Framework, the mechanisms for TPHC operations in Moldova can also be viewed through the lens of the recently developed WHO guiding framework on Multisectoral Action for the prevention and control of NCDs [28]. The framework's four pillars consist of (i) governance and accountability; (ii) leadership; (iii) ways of working; and (iv) resources and capabilities. The TPHC's features and activities are representative of three of those pillars (i), (ii) and (iv). Their multisectoral coordination mechanism, established by the MHLSP and anchored in legislation, sits firmly within the first pillar of the framework, governance and accountability. The second pillar (leadership) is represented by TPHC members networking with professionals through meetings of policy officers across government sectors. Finally, the TPHC's mechanisms for building the relevant knowledge of TPHC members and capacity for multisectoral action falls under the umbrella of the fourth pillar (resources and capabilities). However, while the top-down aspects, i.e., the governance, leadership part and provision of training seem to be formally in place, the bottom-up initiative seems to be lacking to some degree, i.e., the participation in training. To further strengthen the

TPHCs, the following actions are recommended in the areas of governance and accountability, leadership and resources:

## Governance and accountability

- Develop a legal mechanism for the cross-sector approach to public health problems, for the accountability of all the actors involved in solving public health problems, including the mayors' offices, for the formulation of activities aimed at improving population health.

- Update existing regulatory framework with reference to the status and regulation of the TPHC activities, which must go beyond the limits of an order signed by MHLSP. It should offer the possibility to empower representatives of non-medical institutions to participate in meetings and to implement the decisions of TPHC at the government level.

- Draft legal provisions that would regulate TPHC communication and collaboration, and establish a mechanism for dialogue, both horizontally (partnerships at local level) and vertically—with the relevant authorities (MHLSP and Government).

- Revise and specifically de-bureaucratize TPHC nomination procedure and include additional non-medical local institutions (organizations, businesses) that could play a role in influencing public health problems.

## Leadership

- Develop methodological support/procedures at national level for identifying and solving different public health issues through multisectoral interventions and mechanisms for their development, including sharing best practices and experiences at national and international level.

- Develop methodological support regarding the monitoring of the implementation indicators.

## Resources

- Develop an extensive training course for TPHC members in the field of population health, public health problems and possible cross-sector interventions for solving them.

- Strengthen capacities of TPHC members on HiAP and the formulation and monitoring of health programs, based on objectives focused on results and setting monitoring indicators for their implementation.

- Train TPHC members in communicating for a behaviour change with regard to health risks at the community level by presenting health problems and identifying cost-effective solutions.

## Conclusion

The implementation of HiAP at the local level in Moldova was found to be suboptimal with insufficient capacity at all levels to further the aim of reducing health inequalities and achieving its 2030 UHC and other health related goals. TPHC members' ability to deal with public health issues were severely impaired by the general lack of knowledge of those issues or tackling them and understanding how to utilize the TPHC platform for the maximum benefit. Extensive

training for all actors involved in TPHC meetings should better help them understand public health issues and their respective roles as intended by the regulations. This is extended to the chairs of the councils as strong leadership is essential to guide and facilitate intersectoral collaborations horizontally and vertically to overcome the presented public health problems. Incentives for non-health sector members should be incorporated by, e.g., supporting their particular agenda in a way that impacts health to achieve a common goal. Legal reforms of the TPHC framework should consider a common fund for joint ventures, the convening of TPHC members for the council to be engaged and fully effective and ensuring the council's decisions are more binding.

## Supporting information

**S1 File. Figs 1–5.**
(DOCX)

**S2 File. Questionnaire: Territorial Public Health Council.**
(DOCX)

**S3 File. Interview guide.**
(DOCX)

**S4 File. Data and statistics.**
(XLSX)

**S5 File. Perceptions of importance of members for TPHC activity.**
(DOCX)

## Acknowledgments

The authors wish to thank the Moldovan Government, in particular the Ministry of Health, and all those who gave their time to take part and work on a daily basis in the TPHCs.

## Author Contributions

**Conceptualization:** Oleg Lozan, Constantin Rîmiş, Helen Prytherch.

**Data curation:** Valentin Mîţa.

**Investigation:** Oleg Lozan, Valentin Mîţa, Tatjana Buzeti, Peter Beznec.

**Methodology:** Valentin Mîţa, Tatjana Buzeti, Peter Beznec, Constantin Rîmiş, Helen Prytherch.

**Project administration:** Oleg Lozan.

**Supervision:** Ala Curteanu, Robert Canavan, Helen Prytherch.

**Validation:** Daniela Demişcan, Tatjana Buzeti, Peter Beznec, Valeriu Sava, Ala Curteanu, Constantin Rîmiş.

**Writing – original draft:** Robert Canavan.

**Writing – review & editing:** Oleg Lozan, Daniela Demişcan, Valeriu Sava.

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
