## [Decision Letter · Decision Letter 0]

25 Mar 2024

PONE-D-23-35754Assessing capacities to strengthen intersectoral collaboration in Territorial Public Health Councils in the Republic of MoldovaPLOS ONE

Dear Dr. Prytherch,

Thank you for submitting your manuscript to PLOS ONE. After careful consideration, we feel that it has merit but does not fully meet PLOS ONE’s publication criteria as it currently stands. Therefore, we invite you to submit a revised version of the manuscript that addresses the points raised during the review process.

A rebuttal letter that responds to each point raised by the academic editor and reviewer(s). You should upload this letter as a separate file labeled 'Response to Reviewers'.A marked-up copy of your manuscript that highlights changes made to the original version. You should upload this as a separate file labeled 'Revised Manuscript with Track Changes'.An unmarked version of your revised paper without tracked changes. You should upload this as a separate file labeled 'Manuscript'.Please consider reviewer 2's suggestion to improve the discussion as soon as possible.

We look forward to receiving your revised manuscript.

Kind regards,

André Luis C Ramalho, PhD

Academic Editor

PLOS ONE

Journal Requirements:

2. In the ethics statement in the Methods, you have specified that verbal consent was obtained. Please provide additional details regarding how this consent was documented and witnessed, and state whether this was approved by the IRB.

Reviewers' comments:

Reviewer's Responses to Questions

**Comments to the Author**

1. Is the manuscript technically sound, and do the data support the conclusions?

Reviewer #1: Yes

Reviewer #2: Yes

2. Has the statistical analysis been performed appropriately and rigorously? 

Reviewer #1: Yes

Reviewer #2: Yes

3. Have the authors made all data underlying the findings in their manuscript fully available?

Reviewer #1: Yes

Reviewer #2: Yes

4. Is the manuscript presented in an intelligible fashion and written in standard English?

Reviewer #1: Yes

Reviewer #2: Yes

5. Review Comments to the Author

Reviewer #1: After careful consideration (complete reading of the entire manuscript and analysis of the data collection instrument and all graphs and tables in the supplementary file), I believe the manuscript is scientifically suitable for the journal.

Reviewer #2: The article is well written and identifies the situation and needs. To improve the content and messages of the article, it is recommended to use the following framework in the discussion.

This is only a recommendation and important parts of the framework can be used in the discussion.

WHO Framework on Multi-sectoral Action for the Prevention and Control of NCDs and Mental Health

This framework includes the following parts:

1. Governance and Accountability:

• Seeking a mandate, endorsement or supportive legislation for MSA

• Leveraging existing cross-sectoral policies or plans

• Establishing multi-sectoral coordination mechanisms

• Ensuring public accountability

• Developing reporting structures and accountability measures

2. Leadership at all levels:

• Networking with professionals

• Identifying champions

• Establishing incentives or recognition (for/of good multi-sectoral collaboration)

• Setting standards for MSA

• Acknowledging the commitments of other sectors

3. Ways of working:

• Developing communication tools, processes, or activities (to build trust)

• Implementing formal/informal activities that nurture relationship-building

• Establishing knowledge-collaboration activities

• Including diverse stakeholders

4. Resources and capabilities:

• Having dedicated personnel working on MSA strategies

• Encouraging dedicated funding to support MSAs on NCDs

• Strategically building capacity.

Using this framework in the discussion of the article will convey clear messages to the policy makers. If you can't find the framework online, please email me and I will send it to you. ah.bakhtyari@gmail.com

If you have a clear policy recommendation that results from your findings, please briefly mention it at the end

6. PLOS authors have the option to publish the peer review history of their article (what does this mean?). If published, this will include your full peer review and any attached files.

Reviewer #1: **Yes: **Abel Silva de Meneses

Reviewer #2: **Yes: **Ahad Bakhtiari

---

## [Author Response · Author response to Decision Letter 0]

29 Apr 2024

André Luis C Ramalho, PhD

Academic Editor

PLOS ONE

10 April 2024

Dear Dr Ramalho

Re: [PONE-D-23-35754] - [EMID:1f92bb5bc2797b97] ‘Assessing capacities to strengthen intersectoral collaboration in Territorial Public Health Councils in the Republic of Moldova,’

I am hereby referring to your feedback sent to me via email on the 26 March 2024. On behalf of all authors, I would like to thank you for inviting us to address the reviewers’ comments, and I would also like to thank the reviewers for their valuable input and their positive assessment of our work. All issues raised by the editor have been addressed and the reviewer recommendation regarding the discussion has been adopted during the revision to improve the quality of this article. 

I have uploaded two copies of the manuscript. A CLEAN copy and a TRACK CHANGES copy for your disposal. 

Please find below a point-by-point response to the editor’s and the reviewer’s comments indicating the corresponding changes in the manuscript. We hope that the revised manuscript is now suitable for publication in PLOS ONE and look we forward to hearing from you.

Yours sincerely

Helen Prytherch

Swiss Tropical and Public Health Institute

Kreuzstrasse 2, CH-4123 Allschwil, Switzerland

Phone : +41 61 284 8686 / E-Mail : helen.prytherch@swisstph.ch

 

EDITOR

Comment 1: Please ensure that your manuscript meets PLOS ONE's style requirements, including those for file naming. The PLOS ONE style templates can be found at https://journals.plos.org/plosone/s/file?id=wjVg/PLOSOne_formatting_sample_main_body.pdf and https://journals.plos.org/plosone/s/file?id=ba62/PLOSOne_formatting_sample_t itle_authors_affiliations.pdf

Response: We have revisited the style requirements of PLOS ONE and have further aligned the article editing style, namely the size of the headings, the Figure and Table headings and references. We have also uploaded Figures 1-4 in the appropriate format (TIF) and the supplement in PDF. 

Comment 2: In the ethics statement in the Methods, you have specified that verbal consent was obtained. Please provide additional details regarding how this consent was documented and witnessed, and state whether this was approved by the IRB.

Response: When a given participant was identified and recruited, he or she was fully informed about the evaluation. After the study team received a participant’s verbal consent to take part in the study, it was considered that the evaluation tool that was filled in together with the participants was sufficient documentation of their consent. Regarding the IRB, we have stated since the initial submission in the “Ethics” section in the Methods, that “The data were collected according to MoH Provision no. 98d of March 28, 2019, "Regarding the evaluation of the activity of the Territorial Public Health Councils" and “at the time of the study, approval by the Ethics Committee was not mandatory for this type of research.”

Comment 3: Please review your reference list to ensure that it is complete and correct. If you have cited papers that have been retracted, please include the rationale for doing so in the manuscript text, or remove these references and replace them with relevant current references. Any changes to the reference list should be mentioned in the rebuttal letter that accompanies your revised manuscript. If you need to cite a retracted article, indicate the article’s retracted status in the References list and also include a citation and full reference for the retraction notice.

Response: We have reviewed the references and ensured completeness and correctness. We have added one reference in the discussion in response to the recommendation made by reviewer two (reference 28. World Health Organization. Global mapping report on multisectoral actions to strengthen the prevention and control of noncommunicable diseases and mental health conditions: experiences from around the world. 2023). 

REVIEWER 1

General comment: After careful consideration (complete reading of the entire manuscript and analysis of the data collection instrument and all graphs and tables in the supplementary file), I believe the manuscript is scientifically suitable for the journal.

Response: We thank the reviewer for a positive summary of the work presented.

REVIEWER 2

General comment: The article is well written and identifies the situation and needs. To improve the content and messages of the article, it is recommended to use the following framework in the discussion. This is only a recommendation and important parts of the framework can be used in the discussion. WHO Framework on Multi-sectoral Action for the Prevention and Control of NCDs and Mental Health. [Framework details were given here by Reviewer 2]. Using this framework in the discussion of the article will convey clear messages to the policy makers. If you can't find the framework online, please email me and I will send it to you. ah.bakhtyari@gmail.com. If you have a clear policy recommendation that results from your findings, please briefly mention it at the end

Response: We thank the reviewer for their positive assessment of our article and for the valuable suggestion made regarding the inclusion of the recent WHO NCD framework in the discussion. We have followed the suggestion and have added a section to describe the TPHC case example through the lens of the WHO NCD framework, since there is a large overlap of elements relevant to a successful multisectoral action described in the framework and presented by the TPHC structure. We have also added a set of recommendations for strengthening the TPHC structure, again through the WHO NCD framework lens (Lines 675 – 720 in the clean document).

---

## [Editor Report · Decision Letter 1]

2 May 2024

Assessing capacities to strengthen intersectoral collaboration in Territorial Public Health Councils in the Republic of Moldova

PONE-D-23-35754R1

Dear Dr. Prytherch,

We’re pleased to inform you that your manuscript has been judged scientifically suitable for publication and will be formally accepted for publication once it meets all outstanding technical requirements.

Kind regards,

André Luis C Ramalho, PhD

Academic Editor

PLOS ONE
---

## [Editor Report · Acceptance letter]

9 May 2024

PONE-D-23-35754R1 

PLOS ONE

Dear Dr. Prytherch, 

I'm pleased to inform you that your manuscript has been deemed suitable for publication in PLOS ONE. Congratulations! Your manuscript is now being handed over to our production team.

Kind regards, 

on behalf of

Prof. Dr. André Luis C Ramalho 

Academic Editor

PLOS ONE